



# Influence of concurrence of extreme drought and heat events on carbon and energy fluxes in dominant ecosystems in the Pacific Northwest region

Hyojung Kwon, Whitney Creason, Beverly E. Law, Christopher J. Still, and Chad Hanson

Department of Forest Ecosystems and Society, Oregon State University, Corvallis, Oregon, 97330, USA

*Correspondence to*: Hyojung Kwon (hyojung.kwon@oregonstate.edu)

**Abstract.** The impacts of drought intensity and vapor pressure deficit (VPD) beyond historic norms in Pacific Northwest (PNW), USA, are critical in understanding the potential future function and resilience of ecosystems in the

region. While ecosystems in this region are adapted to seasonal droughts, June 2015 temperatures were the highest recorded in the region and strongly coupled with relatively low soil moisture. June is usually the best month for growth in the PNW. Here, we examined the impact of the June 2015 climate extremes on carbon and energy fluxes at sagebrush in the high desert, young and mature ponderosa pine in the semi-arid Great Basin, and Douglas-fir in the mesic ecoregion compared to an average climate year (2014). We assessed if the ecosystems recover from extreme

climate stress within the growing season. The monthly anomalies in temperature and VPD were 3 standard deviations, and precipitation was 1 standard deviation, outside the 30-year mean at all sites. In sagebrush, the carry-over effect of precipitation (i.e., intensive precipitation prior to the drought and heat) mitigated the immediate impact of extreme climate stress, leading to 25-40% increase in net ecosystem production (NEP) and gross primary production (GPP), with little change in ecosystem respiration (RE) and 65% increase in latent heat flux, compared to the June 2014. The

drought and heat lowered NEP by 35-65% and GPP by 15-33% in ponderosa pine and Douglas-fir. A greater increase in latent heat flux was observed in Douglas-fir (110%) than in ponderosa pine (<10%) driven by increased evaporation. The decline in NEP was significantly correlated with VPD ($R^2$ of 0.4-0.7 and p<0.001), but not with soil moisture (0-100 cm). NEP recovered in October (the beginning of the rainy season) in ponderosa pine following the lowest NEP in August (the driest and hottest month). Douglas-fir showed partial recovery by October, resulting in the largest

seasonal reduction (May – October) in carbon fluxes (64-128 g C m$^{-2}$ season$^{-1}$). The seasonal changes in Douglas-fir were corresponding with 30-40% decline and were greater than in mature ponderosa pine (5-20% decline), when compared to the long-term seasonal means (2002-2015), respectively. Our results suggest that the responses of carbon and energy fluxes to climate extremes differ depending on site- and species-specific characteristics. Douglas-fir is likely to experience more constraints on carbon fluxes than ponderosa pine if the hot and dry season intensifies in the

PNW. Given the likelihood of future drought and heat extremes, identifying these anomalous ecological responses to anomalous climate (e.g., the combination of VPD, heat, and dry soil) is critical to improve predictions of physiological thresholds and tolerance of different tree species.



## 1 Introduction

The frequency of extreme climate events, such as drought and heat waves, is expected to rise substantially in the 21$^{st}$ century (Mazdiyasni and AghaKouchak, 2015; Meehl and Tebalid, 2004; Trenberth et al., 2013). An increase in the intensity and duration of these events may impact land-atmosphere interactions in some areas, altering carbon and energy processes of terrestrial ecosystems (Ciais et al., 2005; Gorsel et al., 2016; Schwalm et al., 2012; Teuling et al., 2010; Wolf et al., 2016). Land-atmosphere feedbacks are not uniform and are often complex due to the spatial and

temporal heterogeneity of climate conditions as well as site- and species-specific ecosystem responses to climate variation (Ciais et al., 2005; Schwalm et al., 2012; Teuling et al., 2010). Terrestrial ecosystems respond more substantially to climate extremes than to a gradual change in average conditions. Therefore, it is critical to understand the effect of these events in order to predict the impact on terrestrial ecosystems (Hartmann et al., 2018).

The 2003 European heatwave provides an example of the complex and non-uniform responses of ecosystems to
climate extremes. Europe experienced record-breaking drought and heat events with annual precipitation deficits that were 50% below the long-term mean and summer heatwaves 6 ºC above the long-term mean in July (Ciais et al., 2005; Schär et al., 2004; Teuling et al., 2010). Ciais et al. (2005) reported different reductions of primary productivity and contrasting driving factors (precipitation deficit vs. extreme heat) among sites across Europe. Teuling et al. (2010) found contrasting responses of forest and grassland energy exchange to heatwaves in the central-western Europe. Due
to their more conservative water use, forests exhibited a much larger increase in sensible heat flux than grasslands. Water-limited woodland and energy-limited forest ecosystems responded differently to the 2012/2013 summer heatwave that occurred in Australia (Gorsel et al., 2016). Sensible heat flux increased and carbon uptake decreased in the woodlands while the forest exhibited the opposite trends. These studies highlight a disproportionate and non-linear effect of climate extremes on ecosystem responses, depending on ecosystem type and resource availability in climatic
regions.

The year 2015 is the hottest year on record globally, pushing the global average temperature to at least 1 °C above pre-industrial levels (Tollefson, 2015). The Pacific Northwest (PNW) region of the United States, particularly Washington and Oregon, experienced extremely high temperatures and severe drought in the summer of 2015 (Dalton et al., 2017; Philip et al., 2017). Temperatures in June were the highest ever recorded in the region and strongly coupled
with relatively low soil moisture compared with average moisture in June. Extreme events like those in June 2015 are expected to occur once every 350-400 (Philip et al., 2017).

The PNW has some of the world's most productive forests and consists of a range of ecosystem types across a wide variety of climates from coastal rainforests to semi-arid and arid woodlands and shrublands (Law and Waring, 2014; Littell et al., 2010; Ohmann and Spies, 1998). Due to the different drought and heat sensitivities of these ecosystems
(Falk et al., 2008; Kwon et al., 2018), factors associated with ecosystem responses to extreme climatic variability are likely to differ across the climate gradient. An assessment of the effects of 2015 climate conditions on carbon and energy processes over various ecosystems across the climate gradient will provide insights to ecosystem resilience to extreme events.

Previous studies have primarily examined ecosystem responses to seasonal droughts, focusing on carbon and energy
fluxes and their controlling mechanisms in the region (Falk et al., 2008; Irvine et al., 2004; Kwon et al., 2018; Law et



al., 2001; Ruehr et al., 2012; Thomas et al., 2009; Vickers et al., 2012). Carbon uptake was strongly affected by dry-season precipitation and temperature in young and mature ponderosa pine forests and old-growth Douglas-fir (Falk et al., 2008; Thomas et al., 2009; Vickers et al., 2012). Thomas et al. (2009) found that a reduction in water year precipitation (40%) impacted annual carbon uptake in mature ponderosa pine, showing 44% and 15% declines in net

ecosystem production (NEP) and gross primary production (GPP), respectively, during a severe drought year, compared to the 7-year average. They also found that multiyear drought had a more severe and fundamentally different impact on carbon flux than a single-year drought at the mature ponderosa pine (i.e., 40% reduction in GPP in multiyear drought vs 25% in a single-year drought). Carbon uptake rates decreased with increasing temperature during a drought year within the old-growth Douglas-fir forest (i.e., more carbon loss of 280 g C m$^{-2}$ yr$^{-1}$ in GPP compared to the 6-

year average; Falk et al., 2008). Irvine et al. (2004) observed decreased levels of transpiration in young, mature, and old ponderosa pine under seasonal drought with a more significant decline in young trees than mature and old trees. Although these studies highlight the importance of understanding dissimilar responses of ecosystems, forests such as Douglas-fir and ponderosa pine have received the most attention due to their ecological and economical importance in the PNW. There has been little research in the sagebrush-steppe ecosystem despite its extensive area coverage in

the Northern Great Basin and significance as an ecological and economic resource in the region (Gilmanov et al., 2004; Wrobleski and Kauffman, 2003). Furthermore, none of the events previously studied were as severe as the concurrent drought and heat (used interchangeably with "hotter drought" following Allen et al., 2015) that took place in the June 2015.

In this study, we used eddy flux measurements over multiple ecosystems (sagebrush in the high desert of Eastern

Oregon, young and mature ponderosa pine in the semi-arid Great Basin of Central Oregon, and Douglas-fir in the mesic ecoregion of Southwestern Washington) in the PNW spanning a dynamic range of water availability and temperature in 2015. The measurements across large spatial extents provide a unique opportunity to compare the responses of the four ecosystems to climate extremes. Our objectives are to 1) quantify how extreme drought and heat events during the 2015 growing season influence carbon and energy processes across various ecosystems and climates

in comparison to the 2014 baseline, 2) assess the interactive effects of drought and heat on the processes, and 3) examine if these ecosystems recover from the extreme climate events within the growing season.

**2. Materials and Methods**

**2.1. Study sites**

We conducted our study at four sites spanning various vegetation types and climates of the Pacific Northwest:

sagebrush in the high desert of Eastern Oregon, young and mature ponderosa pine in the semi-arid Great Basin of Central Oregon, and Douglas-fir in the mesic ecoregion of Southwestern Washington (Fig. S1). The high desert sagebrush site (AmeriFlux site US-Bsg, hereafter referred to as SB) is located about 60 km southwest of Burns, Oregon at 1398 meters above sea level (43.4712 N, 119.6909 W; Table 1). The vegetation is predominately big sagebrush (*Artemisia tridentata*) with occasional rabbitbrush (*Chrysothamnus spp.*) and bunchgrasses (*Festuca spp.*).



The young and mature ponderosa pine sites are part of the AmeriFlux's Metolius core site cluster (http://ameriflux.ornl.gov). The young ponderosa pine site (AmeriFlux site US-Me6, hereafter referred to as YP) is located east of the Cascade Range crest near Sisters, Oregon USA and a 23-year old ponderosa pine plantation at an elevation of 998 m (44.3232 N, 121.6078 W). The mature ponderosa pine site (AmeriFlux site US-Me2, hereafter referred to as MP) is located ~16 km north of YP, and the mean stand age is 69 years at an elevation of 1255 m (44.452

N, 121.557 W). The overstory is composed almost exclusively of ponderosa pine (*Pinus ponderosa Dougl*. ex Laws) at both sites. Further details for site description are in Kwon et al. (2018) and Ruehr et al. (2012).

The Douglas-fir forest (AmeriFlux site US-Wrc, hereafter referred to as DF) is located in the T.T. Munger Research Natural Area, a protected section of the Gifford Pinchot National Forest (45.8205 N, 121.9519 W; see Shaw et al. (2004) for more site description) in southern Washington. The site was co-dominated with Doulas-fir (*Pseudotsuga*

*menziesii* (Mirbel) Franco) and western hemlock (*Tsuga heterophylla* (Raf.) Sarg.). However, Douglas-fir has a significant control over the stand carbon sink since the crown is exposed to the highest light conditions, compared to other canopy species (Thomas and Winner, 2000). The stand age ranges from seedling to 500 years old, and the maximum height of the stand is 60 m (Parker et al., 2004; Wharton et al., 2009).

The region experiences hot, dry summers and wet winters and springs with frequent precipitation. Relative to the other

sites, SB has colder winters and hotter summers with less overall precipitation. DF has mild winters with little snow. The 30-year normal precipitation and temperature are 268 mm and 7.4ºC at SB, 488 mm and 7.6ºC at YP, 536 mm and 7.5ºC at MP, and 2220 mm and 9.0ºC at DF (Table 1). These values were calculated using data from the Parameter–Elevation Regressions on Independent Slopes Model (PRISM Climate Group, Oregon State University, http://prism.oregonstate.edu; 1981 to 2010).

**2.2. Eddy covariance and meteorological measurements**

Eddy covariance (EC) measurements at SB were conducted using a closed-path infrared gas analyzer (LI-7200, LI-COR Inc., Lincoln, NE, USA) and a 3D sonic anemometer (CSAT-3, Campbell Sci., Logan, UT, USA) from a 25 m tower. Meteorological measurements include net radiation ($R_n$; CNR1, Kipp & Zonen, Delft, The Netherlands), photosynthetically active radiation (PAR; PQS-1, Kipp & Zonen), and air temperature ($T_{air}$) and relative humidity

(RH; HMP155A, Vaisala, Helsinki, Finland). Soil heat flux (G) was measured using HFP01SC (Hukseflux Thermal Sensors, Deft, The Netherlands). Soil water content (SWC, depths of 5, 10, 20, 50, and 100 cm) and precipitation were obtained from a weather station operated by National Centers for Environmental Information at National Oceanic and Atmospheric Administration (https://www.ncdc.noaa.gov/crn/station.htm?stationId=1023). The weather station was located 50 m from the EC tower. SWC at 5, 10, and 20 cm was averaged to calculate SWC within the top soil layer

(0-20 cm).

Instrumentation at YP and MP was similar to SB. The EC measurement was conducted at 12 m at YP and 33 m at MP, and both sites used an open-path IRGA (LI-7500, LI-COR Inc.; see Kwon et al., 2018 for more information). SWC was measured at multiple depths (10, 20, 30, 50, and 100 cm at YP and 10, 20, 30, 50, 70, 100, 130 and 160 cm at MP) using EnviroSCAN probes (Sentek Technologies, Stepney, SA, Australia) in three locations. SWC at 10, 20, and

30 cm was averaged to calculate SWC within the top soil layer (0-30 cm).



The EC system at DF consisted of a closed-path IRGA (LI-7000, LI-COR Inc.) and a sonic anemometer (Solent-HS, Gill Instruments, Lymington, England, UK) at a height of 67 m (Wharton et al., 2009). $T_{air}$ and RH (HMP-35C, Vaisala) as well as PAR (190-SB, LI-COR Inc.) were measured at a height of 70 m. SWC was measured at an integrated depth of 0-30 cm, 50 cm, and 100 cm (TDR100, Campbell Scientific Inc.). G was measured at a depth of 7.5 cm below the soil surface (HFT-3.1, REBS, Seattle, WA, USA).

Processing of eddy covariance data and the gap-filling method followed Thomas et al. (2009) at SB, YP, and MP, and by Paw U et al. (2004) and Falk et al. (2005) at DF to calculate carbon and energy fluxes. GPP was calculated as the difference between NEP and ecosystem respiration (RE). Negative NEP indicates a carbon source and a positive NEP a carbon sink.

## 2.3. Ecosystem processes and drivers

We calculated the Bowen ratio ($\beta = H/\lambda E$; here H is sensible heat flux and $\lambda E$ is latent heat flux) to assess the impact of the microclimate and hydrological cycle on energy partitioning. To assess the dependency of carbon and energy fluxes on environmental conditions, canopy conductance ($G_c$) was estimated by rearranging the Penman-Monteith equation in the same way in Kwon et al. (2018). To examine ecophysiological responses to drought and heat, we calculated the sensitivity of canopy conductance to VPD using the relationship ($G_c = b - a \cdot \ln VPD$) under non-limiting

light conditions (PAR $\geq$ 500 µmol m$^{-2}$ s$^{-1}$; Oren et al., 1999). The slope of the regression (a) represents the sensitivity of $G_c$ to lnVPD (-d$G_c$/dlnVPD). We used daytime data (10:00 – 16:00 hr) in this study so that variations in carbon and energy fluxes were more apparent due to higher plant activity. This applies to all analyses and results presented in this study.

We assessed the long-term response of carbon and energy fluxes to temperature anomalies at MP and DF (2002-2015) due to their extended data records. We classified $T_{air}$ over the month of June into different standard deviations (1σ to 3σ) from the 30-year temperature mean in June (Fig. S2). The values of the fluxes in each category of $T_{air}$ was compared to those in the background reference. We selected 2014 as the background year because the monthly anomaly of June 2014 had no significant difference from the 30-year monthly anomaly mean We primarily focused

on the ecosystem responses to climate extremes in June when net carbon uptake is highest due to optimal growth conditions. We then extended our assessment to the growing season in order to examine ecosystem resilience to the climate extremes.

As a site-specific drought indicator, monthly standardized precipitation evapotranspiration indices (SPEI) was estimated using the SPEI package (Beguería and Vicente-Serrano, 2013; Vicente-Serrano et al., 2010) in R (R Core

Team, 2016). Positive SPEI indicates wetter conditions, while negative SPEI indicates drier conditions (see Vicente-Serrano et al. (2010) for more information).

## 3. Results

### 3.1. Environmental conditions



The monthly anomaly in $T_{air}$ was 28 to 36% (4.1 to 4.8ºC) and 50 to 70% (0.6 to 0.7 VPD) higher, respectively, in
June 2015 compared to the June 30-year mean (Fig. 1 and Fig. S3). The monthly anomaly in precipitation was 20-50
mm (83-100%) lower in June 2015. The monthly anomalies in $T_{air}$ and VPD were 3σ, and precipitation was 1σ, outside
the 30-year monthly mean at all sites, suggesting an extreme change in $T_{air}$ and VPD but not in precipitation from the
long-term mean. The June 2014 monthly anomalies in $T_{air}$ and precipitation were similar to the 30-year monthly
anomaly mean. Thus, we used June 2014 as the baseline to represent the long-term mean climate for June. The monthly
anomaly of $T_{air}$ was 8 − 9 times higher and the anomaly of precipitation was 2 times lower in 2015 than the background
year of 2014.

The increase in temperature and decrease in precipitation in June 2015 resulted in substantially different environmental
conditions compared with 2014 (Table 2). $T_{air}$ was 8.0-9.5 ºC higher, and vapor pressure deficit (VPD) was 1.3-1.5
kPa higher in 2015 than 2014. Lower precipitation in 2015 coincided with a more negative SPEI (e.g., < -2 of SPEI)
compared to SPEI in 2014 (> -0.2), indicating a higher degree of drought stress in 2015 at all sites (Fig. S4). Despite
lower precipitation in June 2015 at SB, SWC at all depths was higher in 2015 than 2014 (Fig. S5) due to a substantial
amount of precipitation in May 2015 (87 mm), which was the equivalent of ~30% of annual precipitation. At YP,
SWC was similar between the two years at all depths. SWC at MP was lower and depleted more rapidly in June 2015
than June 2014. As the growing season progressed, SWC became similar between the two years. At DF, SWC was
similar at the beginning of June but reduced faster by mid-month in 2015. SWC at all depths remained lower until the
end of the growing season. YP had the lowest SWC among all sites due to higher porosity and a thin organic layer,
while DF had the highest SWC due to higher precipitation and soil water holding capacity.

The daytime energy balance ratio (EBR = $\frac{\sum H + \lambda E}{\sum R_n - G}$) was 0.60-0.94 in June 2015 and 0.72-0.95 in June 2014 across sites.

The values of EBR were within the general range of reported values (0.53-0.90; Stoy et al., 2013; Wilson et al., 2002).
It should be noted that the EBR did not account for soil and canopy heat storage, and the imbalance was not
conclusively discussed in the current study.

### 3.1.1. Carbon fluxes

SB had 40% increase in net ecosystem production (NEP, 0.8 µmol m$^{-2}$ s$^{-1}$) in 2015, whereas YP, MP, and DF had a
considerable decrease in NEP (35-65%; -4.0 to -2.9 µmol m$^{-2}$ s$^{-1}$; Table 3 and Fig. 2). SB showed no change in RE
but a minor increase (25%) in GPP in 2015. At YP, MP, and DF, RE increased from 0.2 to 1.5 µmol m$^{-2}$ s$^{-1}$, but the
increase in RE was smaller than the decrease in NEP, resulting in 1.4 to 3.4 µmol m$^{-2}$ s$^{-1}$ lower GPP (15-33% decrease)
in 2015 than 2014. SB, YP, and MP remained carbon sinks in 2015 during the daytime, as observed in 2014. In 2015,
SB was the only site that showed an increase in carbon uptake during mid-day due to higher precipitation in May 2015.
Unlike the diurnal carbon sink pattern in 2014, DF switched from carbon sink to carbon source in later part of the
daytime in 2015. Overall, the carry-over effect of precipitation (defined as the impact of past precipitation on current
ecosystem process) was evident on carbon flux at SB despite drought and extreme heat conditions, whereas the
influence of drought and extreme heat on carbon flux was apparent at YP, MP, and DF. Reduction in GPP was driven
by differential drought and heat sensitivities of NEP and RE.



We examined ecosystem recovery from the 2015 drought and heat by comparing monthly mean diurnal variation of
NEP between three months in the growing season. We compared June with August, normally the driest and hottest
month and October, which coincides with the beginning of the rainy season. At SB, there was a gradual decline in
NEP from June to October (Fig. 3). NEP was the lowest in August at YP and MP, but the carbon sink strength
recovered in October. At DF, NEP was similar among the three months. Compared to 2014, NEP in October 2015
increased by 23% at SB and decreased by 12% at DF, suggesting that these sites either completely or partially
recovered from drought and heat stress by October.

### 3.1.2.    Energy fluxes

$R_n$ had small changes in 2015 at SB, YP, and MP with no significant difference ($p < 0.05$) when compared to 2014
(Table 2 & Fig. 4). $R_n$ at DF showed an increase of 74 W m$^{-2}$ in 2015, and $R_n$ between the two years was significantly
different ($p < 0.05$). Compared to 2014, $\lambda E$ increased by 45 W m$^{-2}$ but H decreased by -59 W m$^{-2}$ in 2015 at SB. YP
had a minor increase in $\lambda E$ and H, whereas MP had an increase in $\lambda E$ (23 W m$^{-2}$) but a decrease in H (-90 W m$^{-2}$). At
DF, both $\lambda E$ and H increased by 92 and 47 W m$^{-2}$, respectively, with a higher increase in $\lambda E$. G varied the least (< 15
W m$^{-2}$) in comparison to $\lambda E$ and H. G increased more at the open canopy sites (SB and YP) than the closed canopy
sites (MP and DF) which showed almost no change.

The Bowen ratio ($\beta$) decreased substantially from 5.0 in 2014 to 2.1 in 2015 at SB due to the increasing $\lambda E$ and the
decreasing H (Table 2). Minor changes in $\lambda E$ and H at YP led to similar values (1.9 to 2.0) of $\beta$ between the two years.
$\beta$ in 2015 decreased to below 2014 at MP (from 2.1 to 1.3) and DF (from 4.3 to 2.4). At MP, the decrease in $\beta$ in 2015
was more influenced by a lower H, whereas at DF, the decrease in $\beta$ was more influenced by a higher $\lambda E$. These results
suggest the different and non-uniform response of energy fluxes to drought and heat across the different sites.

### 3.1.3.    Effects of drought and heat conditions on carbon and energy fluxes

VPD affected NEP more than SWC at all sites (Fig. 5 and Table S1; $R^2$ of 0.3 to 0.7 and $p < 0.005$), showing a rapid
decrease in NEP with increasing VPD. The weakest influence of VPD on NEP occurred at SB (NEP = 0.4·VPD − 2.0
and $p < 0.005$), whereas the strongest influence of VPD on NEP was observed at MP (NEP = 1.5·VPD − 5.1 and $p <$
0.001), where leaf area index was high (LAI = 2.9). NEP was lower in 2015 than 2014 at all sites except at SB, where
maximum NEP was similar in June of both years because of precipitation in May 2015. As VPD reached about 3.0
kPa, NEP at YP, MP, and DF switched from a net carbon sink to a source and remained a carbon source above 3.0
kPa in 2015. Despite the difference in carbon uptake strength between the two years, the rate of changing NEP along
with increasing VPD was similar at YP and DF, suggesting that concurrent drought and heat did not weaken the
dependence of NEP on VPD. On the contrary, a more pronounced decline in NEP at MP indicates a higher sensitivity
of NEP to VPD in 2015. The influence of VPD on GPP was significant at SB and DF but not at YP and MP. Due to a
strong connection between $T_{air}$ and VPD, the dependence of carbon processes on $T_{air}$ was nearly identical to that on
VPD. SWC at all depths was correlated with RE and GPP at SB, whereas the other sites showed inconsistency in the
influences of SWC on RE and GPP.



The drier sites (SB and YP) exhibited a positive correlation between SWC and $\lambda E$ in the drought and heat year of 2015 ($R^2$ of 0.2 to 0.6 and $p < 0.05$), whereas the wetter site (DF) showed a negative correlation between SWC and $\lambda E$ ($R^2$ of 0.3 and $p < 0.05$; Table S1). No relationship was observed between SWC and $\lambda E$ at MP. SWC had no significant influence on H at all sites. Air temperature was correlated to $\lambda E$ at SB and DF ($R^2$ of 0.2 to 0.5 and $p < 0.05$) and to H at YP and MP ($R^2$ of 0.2 to 0.3 and $p < 0.05$).

Ecosystem response to drought and heat was examined using the sensitivity of $G_c$ to lnVPD (the slope of the regression, -a; Fig. 6). The sensitivity was similar in June between 2014 and 2015 at SB, YP, and DF, whereas $G_c$ was less sensitive to VPD in 2015 than 2014 at MP. Despite the lower sensitivity in 2015, $G_c$ at MP decreased at a higher rate in response to increasing VPD (5 mm s$^{-1}$ kPa$^{-1}$) compared with the other sites (2 to 3 mm s$^{-1}$ kPa$^{-1}$). The sensitivity of $G_c$ to lnVPD was greater in ponderosa pine than Douglas-fir, providing further evidence that in the water-limited ponderosa pine, VPD was a more important driver of $G_c$ than in the mesic Douglas-fir.

### 3.1.4. Response of carbon and energy fluxes to differences in temperature increase

NEP and GPP decreased along with a greater anomaly in $T_{air}$ at MP (Table 4). However, the change in RE was inconsistent and was less than the changes in NEP and GPP. Increasing $T_{air}$ had a lesser influence on $\lambda E$ than H, and $\lambda E$ showed an inconsistent change. H was consistently lower than the June 2014 base period at all temperature anomalies. At DF, all components of carbon fluxes were higher than the base period when the temperature anomaly was below 2σ. Above 3σ, NEP and GPP were considerably reduced but there was little change in RE. The increase in $\lambda E$ and H became larger with increasing $T_{air}$. Compared to H, a greater increase in $\lambda E$ led a substantial decrease in β, suggesting that as $T_{air}$ increased, more energy went towards $\lambda E$ than H.

## 4. Discussion

### 4.1. The carry-over effect of precipitation on carbon and energy fluxes at sagebrush

Intensive rains in May increased soil water content at top and deeper soils through June 2015 (Fig. S5), which provided additional water for shallow-rooted plants (grasses and forbs) and deep-rooted sagebrush (Bates et al., 2006; Knight, 1994; Miller, 1987). The increased soil water content in June enhanced the strength of the carbon sink and allowed for more available energy to be consumed as latent heat flux despite drought and heat stress (Table 3, Fig. 2, and Fig. 4).

Schwinning et al. (2003) found that the amount of water added to soils in spring and summer was linearly related to that of transpiration when examining the relationship between rainfall event size and rainwater use by grasses and shrub plant community in southern Utah, U.S. Although we cannot quantitatively partition evapotranspiration (converted from latent heat flux) into evaporation and transpiration from the eddy covariance data in this study, we speculate that the increased latent heat flux was likely due to a combination of transpiration from shallow and deep rooted plants and evaporation from the wetter soil. In a sagebrush ecosystem, a precipitation shift to spring/summer has the highest potential to change plant productivity and composition of a sagebrush ecosystem (Bates et al., 2006). In this study, the increased soil water availability enabled the sagebrush to tolerate a hotter drought (e.g., more carbon



uptake), demonstrating a strong influence of spring precipitation on carbon and energy processes in the subsequent dry and hot summer. These results suggest that in a water-limited ecosystem like the sagebrush, the lagged effect of precipitation can mask immediate ecosystem response to extreme climate events, and drought-induced changes in these processes may not be properly estimated (Wei et al., 2013).

Because the carry-over effect of precipitation prevents from an explicit understanding of the impact of hotter drought on carbon and energy fluxes at sagebrush, we did not consider sagebrush in further discussion.

### 4.2. Differential response of carbon fluxes to drought and heat

We found that hotter drought substantially reduced the strength of the carbon sink in the ponderosa pine and Douglas-fir forests by impairing ecophysiological processes (i.e., lowering $G_c$). The reduction in NEP was significantly correlated with temperature and VPD but not with soil moisture (Fig. 5 and Table S1), in contrast to other studies demonstrating water scarcity as a primary driver of decreasing carbon sink strength during drought (Allen et al., 2015; Reichstein et al., 2007; Wolf et al., 2016). RE and GPP exhibited a stronger dependence on temperature and VPD than soil moisture. Under existing water stress in June 2015, it is less likely that lower soil water content further constrains the strength of carbon uptake at these ecosystems.

The effect of VPD on carbon uptake was more prevalent in ponderosa pine than Douglas-fir due to higher vulnerability to water-stress-induced xylem cavitation (Fig. 5 and Fig. 6; Minore, 1979; Stout and Sala, 2003). Drought avoidance mechanisms of ponderosa pine were able to offset this vulnerability through stomatal closure (Stout and Sala, 2003). The sensitivity of $G_c$ to lnVPD in June 2015 declined by 50% at YP and 40% at MP, compared to May 2015 (5.0 mm $s^{-1}$ per kPa at YP and 8.4 $s^{-1}$ per kPa at MP), prior to the drought and heat. However, the sensitivity at DF was similar between the two months (2.4 mm $s^{-1}$ per kPa in May and 2.2 mm $s^{-1}$ per kPa in June). The increased cavitation susceptibility of ponderosa pine roots reduce water transport and quicken stomatal closure, causing greater stomatal sensitivity as drought stress occurs (Alder et al., 1996). Therefore, effective drought-avoiding mechanisms likely led to a larger decrease in canopy conductance in ponderosa pine, compared to Douglas fir.

Mature pine had a greater reduction in NEP with VPD, indicating more severe stress in 2015 than 2014. Soil water content (12% at 0-30 cm, 21% at 50 cm, and 26% at 100 cm) in June 2015 was the lowest observed in June from 2002 to 2015 at MP (unpublished data). We speculate that mature pine, which had a higher buffering capacity for water deficit through tree structural properties (stem capacitance and rooting depth; Stout and Sala, 2003), utilized rapid stomatal closure (i.e., higher inherent water use efficiency, Table 3) to avoid hydraulic failure under high atmospheric evaporative demand. At YP, no appreciable difference in the slope of carbon uptake with VPD between the two years suggests that extreme heat in 2015 did not exacerbate the sensitivity of carbon uptake to VPD when soil water content at different depths was below the critical threshold for xylem cavitation (Fig. S5). Increase in RE with temperature may be explained by increased soil efflux (Irvine et al., 2008; Ruehr et al., 2012). In the open forest like ponderosa pine, shallow soil temperature reached as high as 48°C at YP and 36°C at MP in June 2015. It was speculated that stronger increase in temperature may facilitate decomposition of soil organic compound, leading to an increased soil efflux, which accounted for 70% of ecosystem respiration (Law et al., 2001).

### 4.3. Differential response of energy fluxes to drought and heat





The response of energy fluxes to drought and heat track differences in site environmental conditions. The 3σ change in VPD led to a smaller change in latent heat flux in ponderosa pine than in Douglas-fir (Table 2 and Fig. 4). At the

pine sites, the minimal change of latent heat flux may be due to increased evaporation, which compensated for the reduced contribution of transpiration to evapotranspiration (Irvine et al., 2004; Kwon et al., 2018; Law et al., 2000; Ruehr et al., 2012). Kwon et al. (2018) reported that canopy and stomatal conductance in young and mature ponderosa pines substantially declined as soil water deficit increased in Oregon, U.S., leading to a reduced contribution of transpiration to evapotranspiration. Hydraulic systems of young ponderosa pine and Douglas-fir are more vulnerable

to root embolisms than mature ponderosa pine (Domec et al., 2004), and young ponderosa pine is highly vulnerable to extreme heat that can lead to cavitation and reduced NEP (Goldstein et al., 2000). In our study, ponderosa pine had ~60% lower canopy conductance in 2015 compared with 2014 (Table 3). Canopy conductance at the young pine was 50% lower than the mature pine in 2015. Soil water content (7% at YP and 12% at MP) reached a critical threshold associated with the pines' (10%) vulnerability to xylem cavitation (Goldstein et al., 2000). Under the existing water

stress, stomatal down-regulation increased in the ponderosa pines, resulting in a decrease in NEP (Table 3 and Fig. 5) and thus transpiration, when the coupling between NEP and transpiration was considered. However, evaporation increased by depleting the limited remaining soil water supplies during the drought and heat.

The drought and heat lowered sensible heat fluxes in 2015 at the ponderosa pine sites in response to reduced thermal gradient (i.e., Monin-Obukhov length) and aerodynamic conductance for heat. The values of these variables declined

by ~30 m and 10 mm s$^{-1}$ at YP and ~ 40 m and 20 mm s$^{-1}$ at MP in 2015, compared to 2014. The higher rate of change in these values led more reduction in sensible heat flux in MP than YP. The pattern of decreasing sensible heat flux at the ponderosa pines differed from other ecosystems, which showed an increased sensible heat flux at dry sites (e.g., Mediterranean and temperate woodlands) in Australia during an extreme heat wave event in summer 2012-2013 (Gorsel et al., 2016).

At the Douglas-fir, the increased latent heat flux may be explained by the increased evaporation while transpiration decreased. A large decline in NEP (94% reduction) in 2015 reflected a substantial reduction in transpiration through stomatal closure. The higher temperature, which was strongly connected to atmospheric evaporative demand, restricted tree transpiration but enhanced soil evaporation. Daily daytime-averaged latent heat flux continued to increase from 25 to 94 W m$^{-2}$ at the end of June 2015, indicating that soil water was not limiting evaporation.

Evaporation contributing more to evapotranspiration than transpiration was also observed during a drought stress period in a Douglas-fir forest in Oregon, U.S (Kwon et al., 2018). The minimal increase in sensible heat flux may be due to evaporative cooling due to the increased latent heat flux suppressing further heating and depleting soil moisture in Douglas-fir (Gorsel et al., 2016).

Despite the dissimilar responses of latent heat flux to the hotter drought between ponderosa pine and Douglas-fir, both

forest types exhibited limited tree transpiration and increased soil evaporation. Whan et al. (2015) reported that increased evaporation triggers more soil water depletion, which will cause a positive (enhancing) feedback on temperatures as more energy goes into sensible rather than latent heat flux. In summer dry regions, like those included in this study, soil moisture is expected to decline significantly during summer (Mankin et al., 2017), and this will likely shift energy partitioning even more toward sensible heat flux under future climate that is expected to bring



warmer and drier summers to the PNW (Dalton et al., 2013; Wolf et al., 2016). Our results suggest that even within the same plant functional type (i.e., evergreen needle leaf) and the same species with different ages, the hotter drought-induced change in energy process is not uniform due to different interactive effects of specific site environmental and physiological factors (Wilson et al., 2002).

**4.4. Influence of a short-term drought and heat on carbon fluxes within a season**

On a seasonal basis (May – October), all sites remained a net carbon sink by the end of the growing season in 2015 (Table S2). However, a substantial reduction in seasonal NEP (64 g C m$^{-2}$ season$^{-1}$) at MP and DF in 2015 indicated an effect of the short-term drought and heat on the seasonal carbon sink. NEP reduction in June accounted for ~50% of the seasonal difference in NEP between the two years. The remaining difference was largely explained by a further decline in NEE of 30 g C m$^{-2}$ season$^{-1}$ at MP and DF, which was forced by lower precipitation in August 2015 and

cumulative water stress (220 mm lower precipitation in 2015 than 2014), respectively. At YP, the decreased NEP in June 2015 was counterbalanced by an increased NEP later in the growing season, resulting in no difference in seasonal NEP. RE in June 2015 contributed no seasonal difference in RE at YP and MP, whereas at DF, the contribution of RE later in the growing season primarily drove the seasonal difference (64 g C m$^{-2}$ season$^{-1}$). These differential responses of NEP and RE led to a minor variation in GPP at YP and a substantial reduction in seasonal GPP at MP (56 g C m$^{-2}$

season$^{-1}$) and at DF (128 g C m$^{-2}$ season$^{-1}$) in 2015.

To compare the 2015 hotter drought with long-term observations, we examined the interannual variability in seasonal sums of NEP, GPP, and RE during 2002-2015 at MP and DF. At MP, the seasonal sum of NEP was more than -1σ and GPP near -1σ of the decadal means. The magnitudes were equivalent to 20% and 10% declines from the decadal means, respectively. RE was similar to the decadal mean at MP (Fig. 7). At DF, NEP and GPP were more than -

2σ (40% and 35% declines, respectively), and RE was within -1σ (30% decline) of the decadal mean. Despite the considerable reduction in carbon uptake in 2015, the observed changes at MP seemed to be within the decadal variability. Conversely, DF exhibited the highest variability in NEP and GPP in 2015. On a decadal scale, lower variability in seasonal carbon balance in ponderosa pine may be governed by more phenotypic plasticity than Douglas-fir, which is the primary mechanism of tree acclimation to unpredictable environmental conditions (Bradshaw and

Hardwick, 1989; Marias et al., 2017). Ryan et al. (2000) reported ponderosa pine's remarkable tolerance of extremes given the environment in which it exists. The higher variability observed at DF versus MP indicates that Douglas-fir is more vulnerable to the impact of extreme climate than ponderosa pine (Vernon et al., 2018). Another study on the 2015 drought found similar results in Northern California: the annual tree-growth rate of the two species was similar, even though mature Douglas-fir generally has higher annual tree-growth rate than mature ponderosa pine (Vernon et

al., 2018). Increased drought and heat sensitivity in Douglas-fir may be related to greater leaf area of the species (McDowell and Allen, 2015) and more variable growth patterns (Vernon et al., 2018). These differences in the responses illustrate the challenges of modeling drought-and heat-related tree stress and physiological sensitivity. Ecosystem models, which predict forest response to drought and heat, may need to allow for more phenotypic plasticity in some species in order to improve the knowledge in identifying physiological thresholds of tolerance of

different tree species (Law and Waring, 2014).





Hoover and Rogers (2016) reported that shorter term but extreme pulse-drought causes a greater impact on carbon cycle than chronic but subtle drought. Our results emphasize that, even among the same plant functional types, the magnitude of differential sensitivities of carbon processes to a short-term extreme drought and heat can govern how much carbon is lost to the atmosphere or stored by the ecosystem during a growing season. In addition, intra-seasonal variation in environmental conditions is also important in determining seasonal carbon budget by either dampening or amplifying the influence of the extreme event on carbon processes.

## 5. Conclusion

We studied the influence of extreme drought and heat in June 2015, which exceeded the previously observed seasonal drought and heat events in its severity, on carbon and energy fluxes at the predominant ecosystems in the PNW. Our study demonstrated that ecosystem responses to extreme short-term drought and heat event were diverse and non-linear due to the interactive effects of physiological and environmental factors at the sites even within the same plant functional types and species. Under the drought and heat stress, the increased carbon sink and latent heat flux at sagebrush highlight the importance of precipitation prior to drought and heat that can mitigate their immediate impact on ecosystem responses in a dry region. Regardless of different hydraulic properties of these ecosystems, drought and heat substantially reduced carbon uptake at these forests, resulting in a decrease in gross primary productivity. Effective drought-avoiding mechanisms to the climate stress at ponderosa pine resulted in a higher sensitivity of carbon uptake to the stress than at Douglas-fir. When young and mature ponderosa pine were under extreme drought stress, mature ponderosa pine, which had more buffering capacity for water deficit through tree structural properties than young ponderosa pine, appeared to be more sensitive to the extreme temperature stress. The drought and heat played more significant role in shaping latent heat flux in energy-limited Douglas-fir than water-limited ponderosa pines. Young ponderosa pine, the driest site, showed the least change in latent and sensible heat fluxes.

Climate change has already begun to impact carbon and energy balance of forest ecosystems in the PNW by altering frequency, intensity, and duration of droughts and heat stress (Law and Waring, 2014). The types of extreme climate events like June 2015 are projected to increase in length and severity across the PNW (Mankin et al., 2017; Wise, 2016). Uncertainties related to ecosystem responses to extreme climate events pose a key challenge for current land-surface and ecosystem models to predict future vegetation feedbacks and change to global climate with confidence (Mcdowell et al., 2013; Powell et al., 2013; Sitch et al., 2015). Therefore, under rapidly changing climate, accelerated understanding of the diverse patterns and processes driving dynamic ecosystem responses is needed to address effectively the challenge (Hobbs et al., 2014; Perring et al., 2015; Roman et al., 2015). Our study provides an early glimpse of how these major forests are likely to respond to extreme climate changes in the PNW region, emphasizing that the extreme short-term drought and heat did not create equal impacts on carbon and energy processes in major ecosystems in the PNW. Therefore, it is important to assess differential sensitivities of ecosystems to climate extreme by coordinating physiological and structural observations across a range of species to inform the process models (Law, 2014). This study also demonstrates that forest stress from drought and heat is evident through weakened carbon sink strength and that there will be a potential for causing a positive carbon-climate feedback in response to future climate extreme in the PNW.





**Author contribution**

Bev Law and Chris Still designed the research experiment as PI and Co-PI, respectively. Whitney Creason and Chad Hanson conducted the field measurements. Hyojung Kwon conducted data analysis and prepared the manuscript with
contributions from all co-authors.

**Competing interest**

The authors declare that they have no conflict of interest.

**Acknowledgements**

This research was supported by the U.S. Department of Energy's Office of Science (Grant No. DE-FG02-
06ER64318 and DE-AC02-05CH11231 for the AmeriFlux core site) and the US Department of Energy (Grant DE-SC0012194) and Agriculture and Food Research Initiative of the U.S. Department of Agriculture National Institute of Food and Agriculture (Grants 2013-67003-20652, 2014-67003-22065, and 2014-35100-22066) for our North American Carbon Program studies, "Carbon cycle dynamics within Oregon's urban-suburban-forested-agricultural landscapes". We gratefully acknowledge Andres Schmidt, Christopher Thomas, and Dean Vickers for their
contribution to the project. We thank Sonia Wharton and Ken Bible for the data at the AmeriFlux Wind River Crane Site (US-Wrc), which were obtained from AmeriFlux database (http://ameriflux.lbl.gov/).

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

640



Table 1. Site characteristics at sagebrush steppe (SB), young ponderosa pine (MP), mature ponderosa pine (MP), grass (GR), and Douglas fir (DF). Numbers in parenthesis indicate the measurement year.

| | SB | YP | MP | DF |
|---|---|---|---|---|
| Latitude (º N) | 43.4712 | 44.3233 | 44.4520 | 45.8205 |
| Longitude (º W) | 119.6909 | 121.6078 | 121.557 | 121.9519 |
| Elevation (m) | 1398 | 998 | 1255 | 372 |
| Mean age (years) | - | 23 (2014) | 69 (2015) | 375-500 |
| Mean height (m) | | 7.5 (2015) | 18.5 (2015) | 52 |
| Maximum LAI | - | 1.4 (2015) | 2.9 (2015) | 8.2-9.2 |
| 30-year mean annual temperature (ºC) [†] | 7.4 | 7.6 | 7.5 | 9.0 |
| 30-year mean annual precipitation (mm) [†] | 268 | 488 | 536 | 2220 |
| Climate Koeppen | Bsk | Csb | Csb | Csb |

[†] the 30-year mean (1981 to 2010) was obtained from the PRISM Group at Oregon State University

(http://prismmap.nacse.org/nn/).





Table 2. Information on environmental conditions, energy fluxes, and the Bowen ratio (β) at the sagebrush-steppe (SB), young ponderosa pine (YP), mature ponderosa pine (MP), and Douglas-fir (DF) in June 2014 and 2015. The listed variables are air temperature ($T_{air}$), vapor pressure deficit (VPD), precipitation (P), soil water content (SWC; 0-30 cm), net radiation ($R_n$), latent heat ($\lambda E$), sensible heat (H), ground heat (G), and energy imbalance ($\varepsilon$). P is monthly sum, whereas the rest of the variables is monthly mean.

| | | $T_{air}$ | VPD | P | SWC | $R_n$ | H | $\lambda E$ | G | $\varepsilon$ | $\beta$ |
|---|---|---|---|---|---|---|---|---|---|---|---|
| | | °C | kPa | mm | % | W m$^{-2}$ | W m$^{-2}$ | W m$^{-2}$ | W m$^{-2}$ | W m$^{-2}$ | |
| SB | 2014 | 18.8 | 1.7 | 11.7 | 13 | 547 | 268 | 68 | 29 | 98 | 5.1 |
| | 2015 | 26.9 | 3.0 | 3.8 | 19 | 558 | 208 | 113 | 43 | 143 | 2.1 |
| YP | 2014 | 17.8 | 1.3 | 9.3 | 6 | 522 | 265 | 139 | 31 | 87 | 2.0 |
| | 2015 | 26.4 | 2.8 | 0.0 | 7 | 529 | 248 | 138 | 41 | 101 | 1.9 |
| MP | 2014 | 16.2 | 1.2 | 10.4 | 17 | 584 | 353 | 180 | 17 | 34 | 2.1 |
| | 2015 | 25.2 | 2.5 | 0.0 | 12 | 545 | 263 | 203 | 22 | 56 | 1.3 |
| DF | 2014 | 17.9 | 1.1 | 56.1 | 23 | 508 | 244 | 78 | 5 | 180 | 4.3 |
| | 2015 | 27.4 | 2.6 | 0.0 | 21 | 582 | 291 | 170 | 7 | 113 | 2.4 |



Table 3. Magnitudes of monthly daytime averaged carbon fluxes, inherent water use efficiency (WUE$_i$), and canopy conductance (G$_c$) at the sagebrush-steppe (SB), young ponderosa pine (YP), mature ponderosa pine (MP), and Douglas-fir (DF) in June 2014 and 2015. The listed variables are net ecosystem production (NEP), ecosystem respiration (RE), and gross primary production (GPP).

| | | NEP | RE | GPP | WUE$_i$ | G$_c$ |
|---|---|---|---|---|---|---|
| | | $\mu$mol CO$_2$ m$^{-2}$ s$^{-1}$ | $\mu$mol CO$_2$ m$^{-2}$ s$^{-1}$ | $\mu$mol CO$_2$ m$^{-2}$ s$^{-1}$ | g C kPa per kg H$_2$O | mm s$^{-1}$ |
| SB | 2014 | 2.0 | 1.1 | 3.1 | 3.0 | 1.3 |
| | 2015 | 2.8 | 1.0 | 3.9 | 2.7 | 1.6 |
| YP | 2014 | 6.7 | 3.1 | 9.8 | 2.8 | 3.6 |
| | 2015 | 3.8 | 4.6 | 8.4 | 5.2 | 2.0 |
| MP | 2014 | 11.2 | 6.0 | 17.2 | 3.5 | 7.6 |
| | 2015 | 7.2 | 7.5 | 14.7 | 6.4 | 4.5 |
| DF | 2014 | 5.6 | 5.1 | 10.7 | 3.6 | 3.8 |
| | 2015 | 2.0 | 5.2 | 7.2 | 2.9 | 3.4 |



Table 4. Variations in monthly energy and carbon fluxes across differences in temperature increase at mature ponderosa pine and Douglas-fir, in comparison to the background month of June 2014. Numbers in parenthesis indicate the percent change. Negative values indicate a decrease in that response from 2014 to the year(s) within each temperature range (1-3 standard deviations ($\sigma$) from mean). The listed variables are net radiation ($R_n$), sensible heat flux (H), and latent heat flux ($\lambda E$), the Bowen ratio ($\beta$), net ecosystem production (NEP), ecosystem respiration (RE), and gross primary production (GPP). See Fig. S2 for grouping air temperature.

| Site | Standard deviation of temperature anomaly | $R_n$ | H | $\lambda E$ | $\beta$ | NEP | RE | GPP |
|------|------|------|------|------|------|------|------|------|
| | | W m$^{-2}$ | W m$^{-2}$ | W m$^{-2}$ | | g C m$^{-2}$ mon$^{-1}$ | g C m$^{-2}$ mon$^{-1}$ | g C m$^{-2}$ mon$^{-1}$ |
| MP | $0\leq\sigma\leq1$[a] | -51 (-9%) | -91 (-25%) | -8 (-5%) | -0.4 (-16%) | -16 (-16%) | -2 (-3%) | -18 (-12%) |
| | $1\leq\sigma\leq2$[b] | 14 (2%) | -68 (-19%) | -23 (-13%) | -0.2 (9%) | -11 (-10%) | -5 (-9%) | -16 (-10%) |
| | $3\leq\sigma$[c] | -23 (-4%) | -71 (-20%) | 14 (8%) | -0.6 (-27%) | -24 (-24%) | 7 (13%) | -17 (-11%) |
| DF | $0\leq\sigma\leq1$[d] | -1 (<1%) | -4 (-2%) | 37 (47%) | -1.0 (-32%) | -1 (-1%) | 23 (51%) | 22 (23%) |
| | $1\leq\sigma\leq2$[e] | -4 (-1%) | 27 (11%) | 57 (73%) | -1.1 (-36%) | 17 (34%) | 39 (84%) | 56 (57%) |
| | $3\leq\sigma$[f] | 74 (15%) | 47 (19%) | 92 (118%) | -1.9 (-55%) | -33 (-65%) | 2 (5%) | -31 (-32%) |

[a] Data from the years of 2002, 2004, 2006, 2007, 2009, and 2013 at MP.

[b] Data from the year of 2003 at MP.

[c] Data from the year of 2015 at MP.

[d] Data from the years of 2000, 2002, 2004, 2009, and 2013 at DF.

[e] Data from the years of 2003 and 2006 at DF.

[f] Data from the years of 2015 at DF.





**Figure captions**

Figure 1. Monthly anomalies in (A) precipitation (P), (B) air temperature ($T_{air}$), and (C) vapor pressure deficit (VPD) in June 2014 and 2015 at the sagebrush-steppe (SB), young ponderosa pine (YP), mature ponderosa pine (MP), and Douglas-fir (DF) sites.

Figure 2. Mean diurnal variation of net ecosystem production (NEP), ecosystem respiration (RE), and gross primary production (GPP) in June at the sagebrush-steppe (SB), young ponderosa pine (YP), mature ponderosa pine (MP), and Douglas-fir (DF) sites. Gray line and shade indicate the mean and error, respectively, in 2014 whereas orange line and shade indicate in 2015.

Figure 3. Mean diurnal variation of net ecosystem production (NEP) in June, August, and October in 2015 at the sagebrush-steppe (SB), young ponderosa pine (YP), mature ponderosa pine (MP), and Douglas-fir (DF) sites.

Figure 4. Comparison of energy fluxes (net radiation, $R_n$; latent heat flux, λE; sensible heat flux, H; and ground heat, G) in June 2014 and 2015 at the sagebrush-steppe (SB), young ponderosa pine (YP), mature ponderosa pine (MP), and Douglas-fir (DF) sites. Gray boxplot is for 2014, whereas black boxplot is for 2015. Dots and lines in the boxplot indicate means and medians, respectively. Note different scales on theY-axis.

Figure 5. Comparison of the changes in daily sum of net ecosystem production (NEP) with daily averaged vapor pressure deficit (VPD) between 2014 and 2015 at the sagebrush-steppe (SB), young ponderosa pine (YP), mature ponderosa pine (MP), and Douglas-fir (DF) sites.

Figure 6. Comparison of the sensitivity of canopy conductance ($G_c$) to vapor pressure deficit (VPD) in June between 2014 and 2015 at the sagebrush-steppe (SB), young ponderosa pine (YP), mature ponderosa pine (MP), and Douglas-fir (DF) sites.

Figure 7. Interannual variability of seasonal net ecosystem production (NEP), ecosystem respiration (RE), and gross primary production (GPP) and seasonal anomaly in vapor pressure deficit (VPD) at the mature ponderosa pine (MP) and Douglas-fir (DF) sites. The long-term mean and standard deviation (σ) were estimated using the data from 2002–2015 at MP and DF. Seasonal values were calculated using the data from May to October.




Figure 1

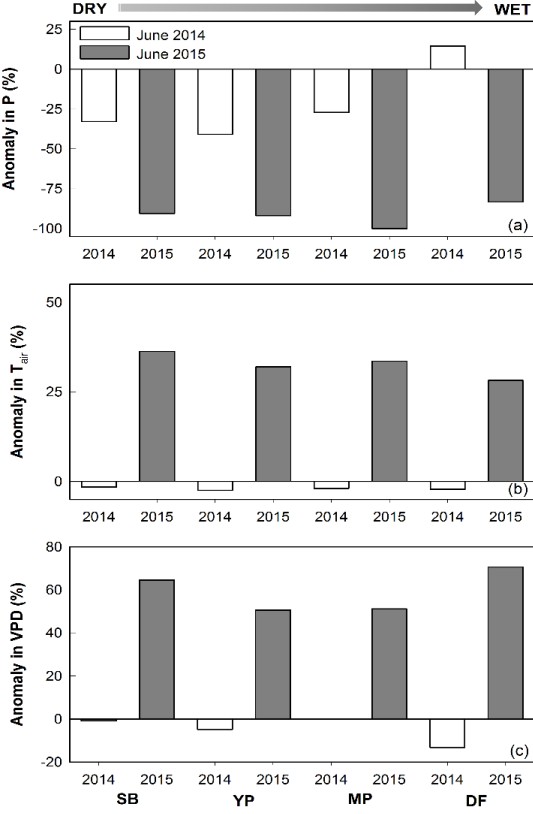





Figure 2

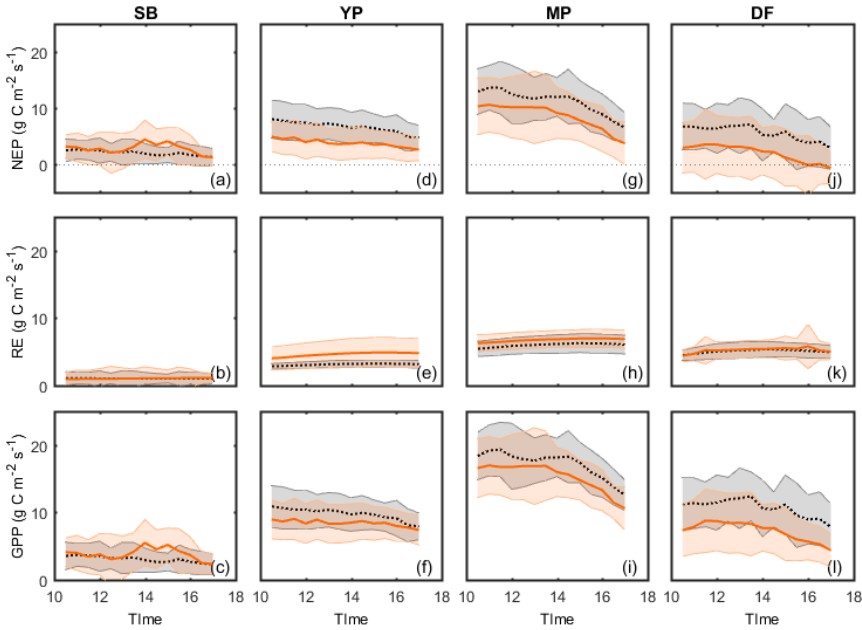





Figure 3

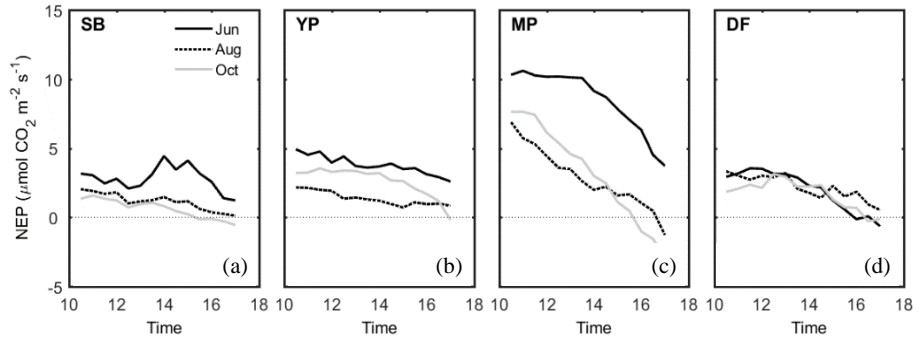



Figure 4

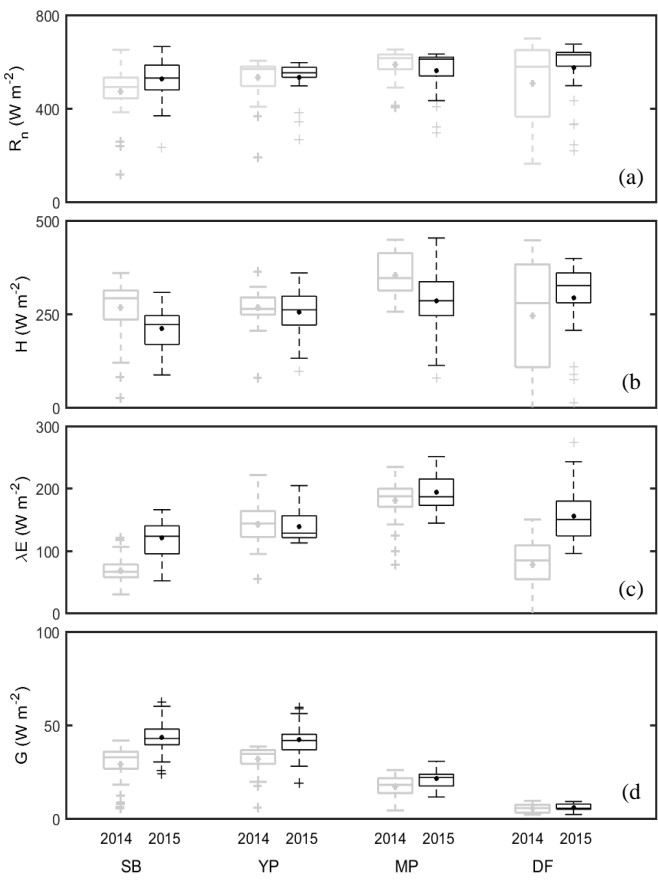





Figure 5

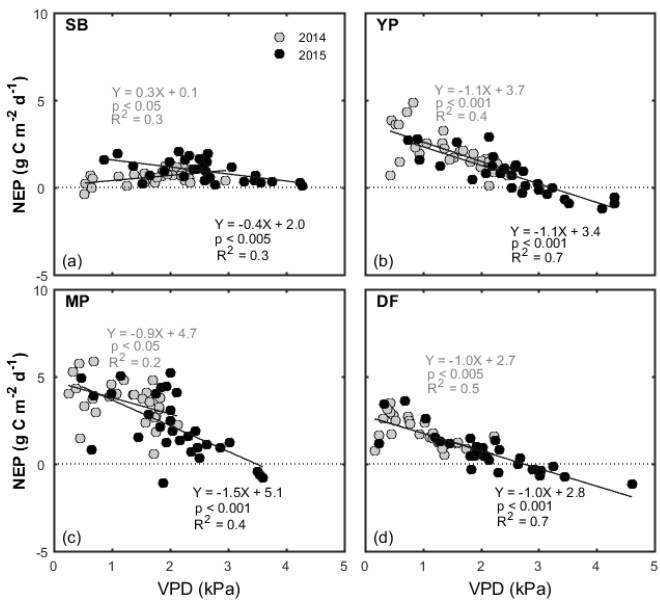



Figure 6

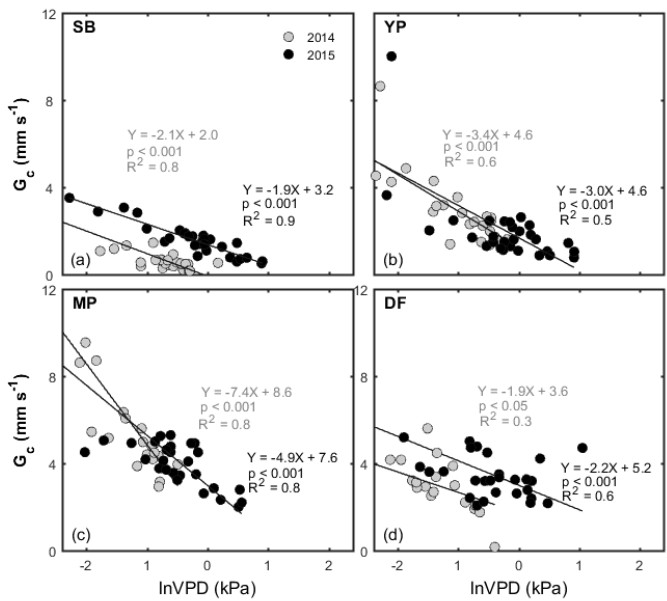





Figure 7

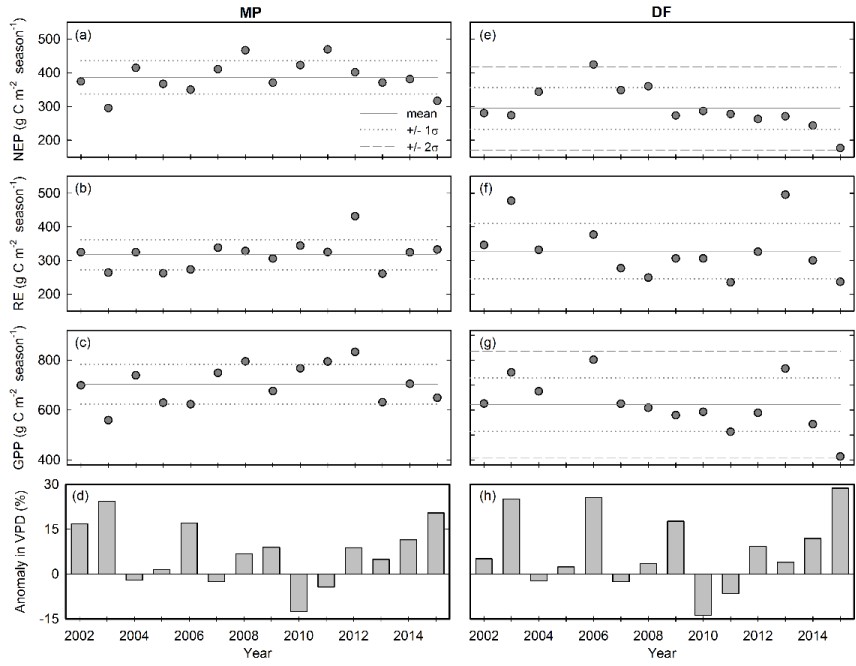