# Peer review of "Influence of concurrence of extreme drought and heat events on carbon and energy fluxes in dominant ecosystems in the Pacific Northwest region"

_Biogeosciences, 2018_

## Referee Comment (RC1) · Anonymous Referee #1 · 5 Aug 2018

General comments

The study by Hyojung Kwon and others took advantage of several sites which represent a natural gradient in climate (mainly precipitation) and vegetation to examine the impacts of an extreme hot and dry event (in June 2015) on carbon and energy exchange in different ecosystems. The authors addressed an important and interesting question, and I agree with the authors that the study system provides a unique opportunity to compare the responses of different ecosystems to the same extreme climatic event. The discussion paper has a clear structure and is generally easy to follow. However, I have a few major concerns on how the authors analyzed and represented their data.

[Figure]

1.Results on carbon and energy fluxes seem largely independent of each other in this paper. If this is the case, why bother integrating these seemingly unrelated results in a single paper instead of writing two independent papers (one for carbon fluxes, the other for energy budgets)? Carbon and energy balance are in fact coupled in ecosystems and feedback to each other via, for example, water use efficiency (WUE). Large fractions of carbon and latent heat (LE) fluxes exchange through stomata. Although the authors showed inherent WUE (WUEi) in Table 3, they did not describe the definition and calculation of WUEi in the Materials and Methods section, nor did they interpret the results related to WUE (or WUEi) in the text. I would suggest that the authors focus more on the coupled responses of CO2 and water vapor (equivalent to LE) fluxes to the extreme event. Examining WUE or WUEi in detail may be a useful way to wrap the results of this paper into a tight story, and could also be valuable in understanding water use strategies of different ecosystems (the authors discussed a lot about water use).

2.The definition of drought should be clarified in this paper because it affects the interpretation of results. For example, if drought is defined as low soil water content, then the observed increases (rather than decreases) in NEP and GPP at the SB site in June 2015 is simply because no drought occurred during that time. Drought can also be defined by precipitation relative to evapotranspiration (the authors used SPEI). Some studies define drought as low VPD. In addition, the concept of "carry-over" effects is more often used at longer timescales, i.e., carry-over from one year (growing-season) to the next. What the authors reported for the SB site, i.e., increases in NEP in June as a result of high precipitation in May, is commonly referred to as the effects of "antecedent conditions".

3.I don't understand how the analyses in the current paper can reveal independent and interactive effects of drought and heat on ecosystem fluxes (objectives 1 and 2 at the end of Introduction). Most of the results are largely descriptive. Moreover, authors examined the relative controls by Ta, VPD, and SWC at different depths using only

simple linear regressions, which may provide unreliable results due to confounding factors or multicollinearity. I would thus encourage authors to think of other analyses (e.g., variability partitioning techniques, multivariate statistics) that could provide more convincing evidence.

4.Responses of ecosystem respiration (RE) to environmental factors is problematic in this study, the authors used only daytime data (10:00-16:00 hr) for all analysis (see Fig. 2). However, the eddy-covariance (EC) technique does not give direct measurements of RE during daytime. Daytime RE is usually extrapolated from nighttime relationships with environmental factors (e.g., temperature and moisture). Therefore, nighttime NEE data should be used when examining RE in response to the extreme event.

5.This study mainly addressed the effects of the extreme event on short-term carbon dynamics (monthly and diurnal changes in carbon sinks and sources, for example, Lines 202-205). I think it is more important to understand how climate extremes could affect ecosystem carbon balance at seasonal and longer timescales. Therefore, Figure 7 (seasonal fluxes) should be in Results instead of Discussion. I also suggest examining to what degree the climate extreme can affect annual budgets of ecosystem fluxes.

Specific comments

1.Line 1, the title may be revised a little bit "Influence of co-occurring drought and heat events on carbon and energy fluxes in dominant ecosystems of the Pacific North-west".

2.Line 10, ecosystems do NOT adapt to droughts, they frequently experience seasonal droughts.

3.Line 15, duration of the growing season?

4.Lines 30-32, delete this sentence or change to another one. This statement is more of justification of the study than of implication of the results.

5.Lines 39-41, I don't see any logic with the preceding or following sentence. Please

consider deleting or revising this sentence.

6.Lines 45-47, deleting this sentence does not hurt.

7.Lines 76-78, this sentence seems irrelevant to this study and should be removed.

8.Lines 71-86, these case studies should be summarized rather than listed. Furthermore, this part should address particularly 1) why this record-breaking event in 2015 may lead to ecosystem responses that had not been observed previously; 2) why and how different ecosystems along the gradient may respond differently to the event?

9.Line 127, what was the height of the sagebrush canopy? I suppose an EC instrument height of 25 m is too high for this type of vegetation.

10.Lines 126, 136, and 141, I noticed that different models of the EC system were used. I understand that although different models (especially open- vs. closed path IRGA) may brought uncertainties into results, the conclusions of this study should be robust because it mainly reported changes in fluxes in relative terms (percentage changes). However, some justifications or clarifications should still be provided in section 2.2.

11.Lines 191-192, please delete this sentence and also remove any other words serve to explain results in the Results section (for example, lines 228, 252-253). Explain results in Discussion, if necessary.

12.Lines 193-196, the energy balance closure and its potential impact on results should be described in the Materials and Methods section.

13.Lines 202-205, consider removing these trivial results. Changes in carbon sink/source over a day are of little importance.

14.Lines 205-208, these words read more like discussions.

15.Lines 209-211, this should be moved to the Materials and Methods section.

16.Line 217 and 219 (also other places), what significance test was used to produce

$p < 0.05$? The corresponding statistics (for example t value and df) should also be provided.

17. Lines 233-234, this was already mentioned in 3.1.1.

18. Line 238 (and also other places), significance tests should be performed to examine whether slopes (sensitives) differ between years or among sites.

19. Section 4.1 should be shortened since it is not surprising to see that higher rainfall in May could enhance water available for production in June.

20. Line 288, this conclusion was derived only from simple linear regressions, and this is not reliable.

21. Line 307, "increase in RE", start a new paragraph here.

22. Line 392, the conclusion is too long and should be condensed.

23. The authors described carry-over effects, responses during the event, and post-event recovery. Therefore, I would suggest adding a timeseries figure spanning these three stages alongside the baseline timeseries (i.e., for 2014). This figure should include both ecosystem fluxes and key environmental drivers to give readers an intuitive visual inspection of what happened during the two years.

24. Table 2-4, which variables are mean daytime values (10:00-16:00?), and which are not? I suppose precipitation sums (Table 2) also include nighttime values. This should be clarified in each table title.

25. Table 3, WUEi was described neither in Materials and Methods nor in Results.

26. Table 4. Variations in monthly energy and carbon fluxes in June 2015 . . .

27. Figure 3, for examining the recovery process, the y-axis should be percentage changes (%) relative to the normal year values of the same month.

28. Figure 4, what are the sources of variations do the error bars and boxes represent?

Why Figure 4 (energy flues) was represented as a boxplot, while Figure 2 (carbon fluxes) as a line plot? Why not using the same type of plot for carbon and energy fluxes?

29.Figure 5, the figure legend tells that daily sums and daily averaged values were shown, while in the MM section (line 157) the authors wrote "We used daytime data (10:00-16:00 hr) in this study". This is confusing. The authors should clarify in both the main text and figure legends (please check all figures) where daytime data were used, and where data for the whole day were used.

Technical corrections

Line 8, . . . historical norms . . .

Line 9, . . . critical to understanding . . .

Line 93, Our objectives were to . . .

Line 157, please provide the time zone.

Table 4, Data from the year of 2015 at DF.

Figure S1, should US-MRf be US-WRC instead?